# Response of Channel Morphology to Climate Change over the Past 2000 Years Using Vertical Boreholes Analysis in Lancang River Headwater in Tibetan Plateau

Yinjun Zhou [1,*,†], Yu Gao [1], Qinjing Shen [2,†], Xia Yan [1], Xiaobin Liu [1], Shuai Zhu [1], Yuansen Lai [3], Zhijing Li [1] and Zhongping Lai [2]

[1] Key Laboratory of River Regulation and Flood Control of Ministry of Water Resources, Changjiang River Scientific Research Institute, Wuhan 430010, China; gygongwu@163.com (Y.G.); yxiawhu@163.com (X.Y.); lxb2005@163.com (X.L.); zhushuai1209@foxmail.com (S.Z.); lzjketty@126.com (Z.L.)

[2] Guangdong Provincial Key Laboratory of Marine Disaster Prediction and Prevention, Institute of Marine Sciences, Shantou University, Shantou 515063, China; shenqinjing@foxmail.com (Q.S.); zhongping.lai@yahoo.com (Z.L.)

[3] Guangdong Province, Shantou, Jinping District, Shantou University Affiliated Middle School, Shantou 515063, China; lai_yuansen@163.com

* Correspondence: zhouyinjun1114@126.com

† These authors contributed equally to this work.

**Abstract:** The Qinghai-Tibetan Plateau, known as the world's "third pole", is home to several large rivers in Asia. Its geomorphology is exceptionally vulnerable to climate change, which has had a significant impact on historical riverbed development through runoff and sedimentation processes. However, there is limited research combining climate change, sedimentology, and chronology with river dynamics to investigate riverbed evolution patterns in geological-historical time scales and their changes in overland flow capacity. In the current study, the evolution of a representative portion of the river channel in the Nangqian basin in the Lancang River headwaters was investigated to explore the reaction of the riverbed to climatic change during the geological period via field surveys, riverbed drilling, optically stimulated luminescence (OSL) dating and bankfull channel geometry parameters. The generalized channel section of the historical period was obtained by linking sedimentary layers of the same age on the distribution map of borehole sections, and the bankfull area of the river was computed accordingly. The restored bankfull areas can effectively reflect the ability of historical river channels to transport water and sediment, thus reflecting the climate change at that time. The findings showed that river morphology in the mounded river section could be successfully reconstructed using OSL dating and sedimentary records and that the conceptual sections of the historical warm periods at 2000 years (2 ka) and 0.7 ka can be recovered. Based on the reconstruction, the calculated bankfull areas during the two warm events were larger than present by factors of 1.28 and 1.9, respectively, indicating a stronger capacity for transporting water and sediments. This is the first trial in the Lancang headwaters to investigate the response of river morphology to climate change on a geological time scale.

**Keywords:** Nangqian basin in Lancang River; channel sediment cores; optically stimulated luminescence (OSL) dating; paleoclimatic changes; river morphology reconstruction; bankfull area

## 1. Introduction

The Lancang River headwater is the source of the Mekong River. Climate change affects the processes of both source and sink, i.e., headwaters and coastal deltas. The Lancang River is the ninth longest river in the world, and its source has received increasing attention in recent years due to research on its runoff, evolutionary processes, and future change under climate change will play a critical role in multi-national policy making in water management [1]. However, because of its location in the hinterland of the Tibetan

Plateau (Figure 1), previous studies are very limited due to difficult access [2]. According to investigations using satellite images, the river channel in the Nangqian basin in the river headwater was taken as a typical reach. The river section is controlled by the backwater of a canyon (nickpoint) just downstream, and river beaches are well developed. The Nangqian basin was filled with sediments, which was ideal for paleochannel reconstruction.

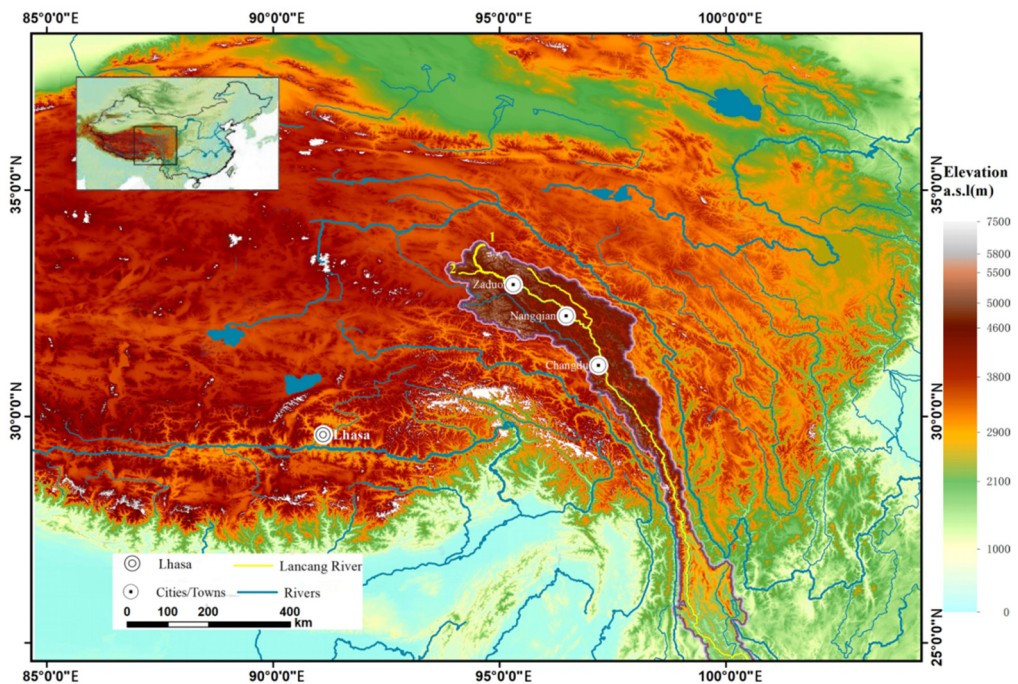

**Figure 1.** Location and elevation of the Lancang River Basin.

The historical river evolution is an important issue for comprehending river geomorphic dynamics, providing a foundation for understanding larger-scale river evolution, and is a fundamental reference for predicting river evolution in the context of climate change. Unlike analyses of modern river evolution, which are based on actual measured information, assessing historical river evolution often lacks reliable information and data. According to previous studies, historically, such analyses have primarily been carried out from river history examinations, in addition to regional tectonic background, paleochannel, and alluvial fan surveys at the mouth of the river. Nash et al. [3] used digital elevation models (DEMS) from the Shuttle Radar Topographic Mission (SRTM) to map the profile of the Kalahari Desert river network, finding that the network development was very sensitive to tectonic movements, from where most of the river valley inflections were derived.

Indeed, most mainstream methods of paleochannel investigations have been conducted via remote sensing interpretation of bend relics. Wang et al. [4] mapped the river banks of the Yinchuan plain section along the Yellow River using MSS data in 1975 (resolution 60 m) and TM data in 1990, 2010 and 2011 (resolution 30 m) to derive the average oscillation rate of the river banks; whereas Yao et al. [5] similarly studied the changes in riverbank erosion and siltation areas in the Ning-Meng section of the Yellow River from 1977 to 2008 via remote sensing analyses, concluding that there was an exponential relationship between flow and the average annual erosion area of river banks, with a non-negligible influence of human activity. Mao et al. [6] used a combination of multi-source remote sensing and topographic data to identify two large-scale river channel changes in the Qingkou area of the Huai'an section of the Beijing-Hangzhou Grand Canal (China), finding that both natural (e.g., the scouring effect of water flow) and human factors (e.g., improvement of navigation channels) have affected local river channel evolution. Arnez et al. [7] examined the factors influencing the evolution of the Ichilo River (Bolivia) level in the Amazon Basin

via both remote sensing analyses and fieldwork, revealing the importance of cut bends, climate, and human activities. Li and Wang [8] analyzed the distribution and causes of river types in the Maqu section of the Yellow River (China) source through a similar combination of remote sensing imagery and field investigations, revealing that the differences in river type distribution were mainly due to topographic constraints, sediment accumulation, changes in bed specificity and riparian material composition, in addition to the river in- and out-flow. Lombardo [9] studied satellite imagery of Amazon creeks from 1984 to 2014, suggesting that creek rift development in the region was not related to El Niño activity; rather, the frequency of rift occurrence was controlled by intra-basin processes on timescales from 1 to 10 years, and the average location of rifting was controlled by climatic or neotectonic events on a millennial scale.

Analyses of riverbed evolution on the centennial scale can be carried out by means of field observations, aerial photography, or satellite imagery; whereas longer-term evolution studies need to combine with sediment and geomorphological features. Cohen and Daley [10,11] examined a 10,000-year-scale river evolution by vertical boreholes on river terraces and river floodplains in eastern Australia. Wu [12] revealed prehistoric mega-flooding by examining a set of special angular debris accumulations, thereby greatly advancing the understanding of evolutionary mechanisms and river histories. Jiang [13] used a high-density resistivity method to detect Holocene paleochannels, deriving a three-dimensional morphology of paleochannels, and their spatial spreading characteristics through a three-dimensional visualization software. Sánchez Vuichard et al. [14] reconstructed the paleoenvironmental history of the Cabeza de Buey shallow lake in Spain over the past 600 years through a multi-proxy analysis that included pollen, non-pollen palynomorphs, and plant macrofossil remains. Lim et al. [15] investigated the variability of flooding recorded in the fluvial floodplain deposits, suggesting that the centennial-scale of flooding variability is associated with climate. Nivière et al. [16] used cosmogenic nuclide [10]Be exposure dating to date an alluvial terrace of the Aspe River in the foothills of the northwestern Pyrenees, pointing out that repeated climatic oscillations induced several cycles of valley filling and subsequent downcutting associated with superimposition, leading to the present-day complex of crosscutting buried drainage networks.

With the development of sediment dating technologies in recent years, research in the field of river terraces has become more attainable. The present study was based on a selection of river sections with terrace development and well-preserved natural river sections, to obtain a dating reconstruction of ancient flood levels and historical sections. Generally, river selection is based on a single type of flushing and silting equilibrium with a decreased influence of human activities. There are many types of dating methods: [14]C, optically stimulated luminescence (OSL), radioactive elements, stratigraphic comparison, event dating, archaeological, and other dating methods are available, among which [14]C represents the most mature technique. However, [14]C dating materials are not well-preserved in river samples, and OSL dating has become an increasingly important dating alternative due to its large measurement time range, high accuracy, and adaptability to samples [17–22].

In recent years, many scholars have carried out extensive research on the relationship between the alluvial plains and climate, based on sediments and the deposition rate using [14]C and OSL [23–25]. Notebaert and Verstraeten [26] integrated the study of river depositional processes in the alluvial plains in central and western Europe since the Holocene, and found that in the early Holocene, most alluvial plains were relatively stable; in the middle and late Holocene, the deposition rate increased and accumulated rapidly since nearly 2 ka. Changes in river deposition rates can reflect anomalous changes in climate [27]. The study of deposition rates in the floodplain of the Morava River since 1300 years has found that changes in deposition rates are related to the diversion and reorganization of rivers caused by extreme climatic events in Central Europe [28].

In the current study, boreholes were obtained following a cross-channel transaction, and riverbed sediments were dated by OSL in order to restore the channel morphology and establish the relationship between riverbed evolution and climate change in the Nangqian

basin of the Lancang River headwater. The size of the bankfull area was calculated based on the reconstructed cross-section channel morphology and runoff. Sediments transport capacity was also estimated.

## 2. Date and Methods

### 2.1. Study Area

The Lancang River originates in Zaduo County, Yushu Prefecture, Qinghai Province, China, and has two sources: Zha'aqu to the north, and Zha'naqu to the west. Although the difference in length between the two sources is relatively minor, the watershed area and water volume of Zha'aqu are significantly greater; whereas Zha'naqu contains a larger amount of sand. After the confluence of these two tributaries, the river is named Zaqu and merges with the left bank tributary Ziqu in Xialaxiu. After flowing southeast to Changdu, Tibet, and merging with the right bank tributary Angqu, the river has the name Lancang River (Figure 2). The source area of the Lancang River is narrowly defined as the area above Zaduo County, Qinghai Province, where the main stream is >200 km long. At present, the Lancang River in the Qinghai Province is also generally considered as the source of the Lancang River, with a main stream length of 454 km, and a watershed area of 37,000 km$^2$.

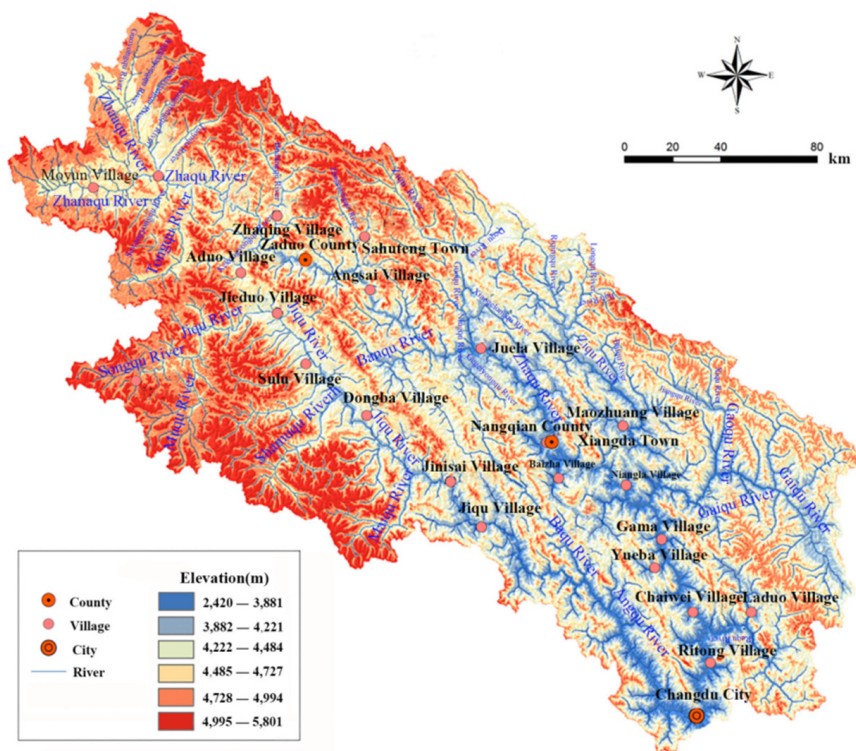

**Figure 2.** Hydrological map of the upper Lancang River.

The river channel in the Nangqian basin is wide and shallow, containing numerous streams and branches, although both banks are bounded by low hills, and the riverbed only oscillates within a certain range of erosion and siltation. Downstream is the Tibetan Canyon, which plays a role in controlling local river section congestion. The surface sediments of the riverbed are primarily sand and gravel, with limited effects from human activities. This section has the largest number of upstream hydrological stations in the Zhaqu–Xiangda section (see Figure 2), which monitors a watershed area of 17,900 km$^2$. The relationship between the water level and flow in this multi-stranded branching river type section is relatively stable, with a multi-year average flow of 138 m$^3 \cdot$s$^{-1}$, a sediment transport volume of 3.55 million tons per year, an average sand content of 0.811 kg$\cdot$m$^{-3}$, and a multi-year average sediment transport modulus of 199 t$\cdot$km$^{-2} \cdot$a$^{-1}$ [1].

### 2.2. Historical River Morphology Recovery in Data-Poor Areas

The mechanisms for the digital recovery of section morphology using sediment stratification, determined by OSL dating, are as follows (Figure 3): First, the alluvial river section is divided into several sections, and several locations are selected for sediment vertical drilling. Sampling is based on a specified transaction. Next, according to the sediments' particle size, the drilled sediment columns are divided into several layers, each of which is dated by OSL. Finally, sediment layers of the same age are connected, and the coordinates are digitized and graphically processed to complete the digital recovery of the historical river section (i.e., the digital recovery of the historical form of the whole river section after integration; Figure 3). Only when the sediment deposition exceeds the erosion, the digital recovery of the historical river section could be recovered. In order to restore the historical morphology of the plateau rivers with alternating wide and narrow channels, this study preferentially selected the wide valley reach with canyons in which sediments could be allowed to accumulate. Specific steps are shown in Figure 4.

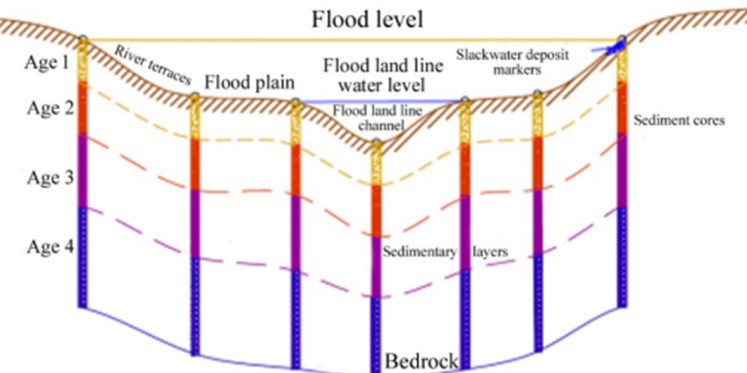

**Figure 3.** Conceptual diagram showing the restoration of historical river cross-sections.

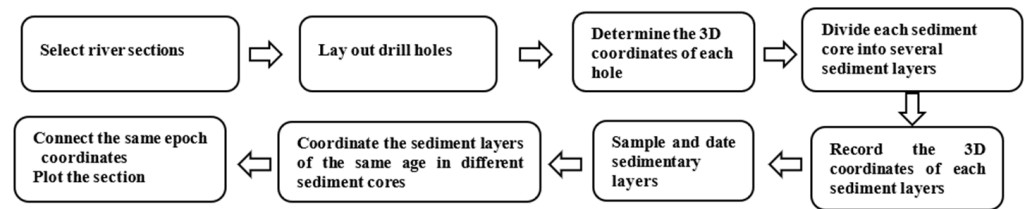

**Figure 4.** Conceptual flow chart of historical morphological cross-section recovery in the Lancang River.

### 2.3. Sample Collection

A HZ-130Y core drilling rig was used to drill the riverbed sediment cores along the cross-section of the Nangqian River, with a hole diameter of 10 cm and a drilling depth of 3–5 m. Considering the transportation conditions, operating space, and avoiding obvious areas of human activity, six sections with side beaches and first-order terraces were selected for drilling (the layout of locations is displayed in Figure 5, and all sections are displayed in Figure 6). The basic information on the river type and the number and depth of boreholes in each section are shown in Table 1. Notably, the N4 section had the most stable beach development because of the upstream control of paired nodes and water flowing into this channel. To locate the borehole position, the methods used for field measurement were based on a hydrographic survey, sample collection, and measurement, in addition to image data collection. River width was measured directly by a laser range telescope, riverbed elevation was located by GPS, and water depth was estimated by scale or field reference.

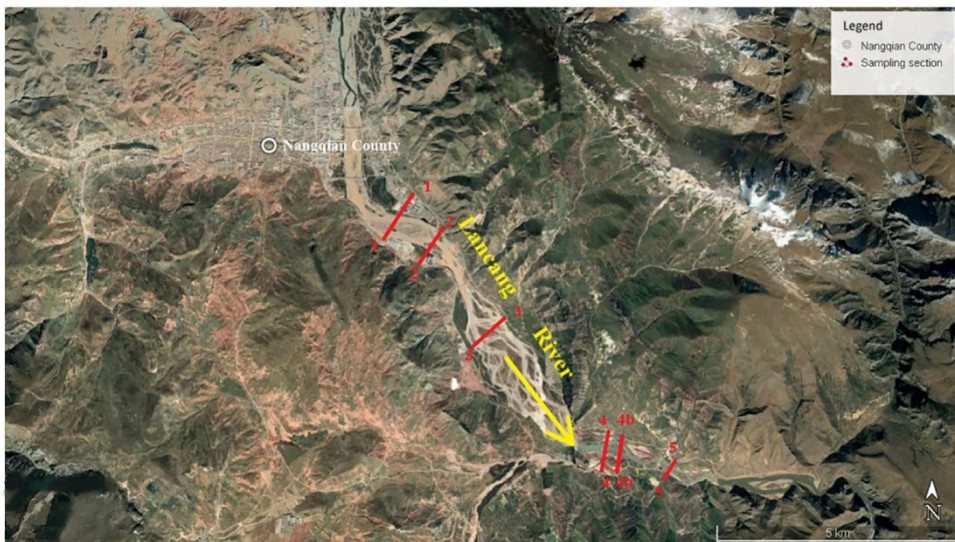

**Figure 5.** Sampling section distribution along the Lancang River source. (The direction of the arrow is the direction of the river flowing).

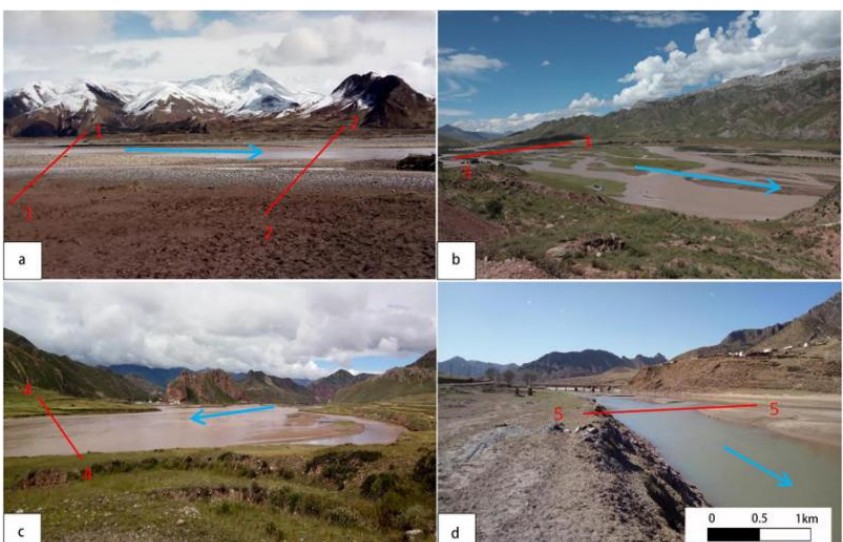

**Figure 6.** Photographs of field geomorphology and river flow direction for the sampled river sections along the Lancang River source. (**a**) shows Section 1 and Section 2; (**b**) shows; (**c**) shows Section 4; (**d**) shows Section 5. The direction of the arrow is the direction of the river flowing).

*2.4. Luminescence Dating*

2.4.1. Luminescence Sample Collection and Pre-Treatment

A total of 30 OSL samples were collected from two sections, N4 and N4b. Based on sample lithology, particle size, and preservation, 16 samples were selected for dating. Sample lithology and burial depth are shown in Figure 7. All samples were collected based on the following two requirements: (1) Boreholes were drilled at locations with a natural sedimentary environment, far from artificially disturbed areas to avoid the influence of local sand panning and farming; (2) selected borehole locations were representative, and their geomorphology was consistent with the major geomorphologies of nearby areas, including riverbeds, terraces, and river floodplains; thus better to be able to reflect the sedimentary and chronological conditions of a certain geographic range.

**Table 1.** Description of river type and boreholes in each section.

| Profiles | Maximum Drilling Depth | No. of Boreholes | River Type | Human Activity along Riversides |
|---|---|---|---|---|
| N1 | 3.7 m | 8 | River maintains three branches. The rate of widening is 5.7%. | Near the county with more human activity: |
| N2 | 4.0 m | 4 | River has a relatively low water level, two branches, and a 3.8% rate of widening. River channel is ≤800 m wide. | Traces of sand mining activities on both sides of the river. |
| N3 | 4.2 m | 8 | River has relatively low water level, two branches, and a 3.8% rate of widening. River channel is ≤ 1000 m wide. | No obvious human activity. |
| N4 | 5.0 m | 7 | River is in a single form, with a relatively low water level and slight curve. There are bedrocks protruding from both sides of the upstream. | The beaches on both the left and right banks are in a quasi-natural state. |
| N4b | 4.2 m | 4 | River is in a single form, with a relatively low water level and slight curve. The river channel is ≤800 m wide. | No human activity except from upland roads. |
| N5 | 5.1 m | 4 | Straight single river type. River width is ~210 m, gradually decreasing into the canyon section. | No obvious human activity on the beach. |

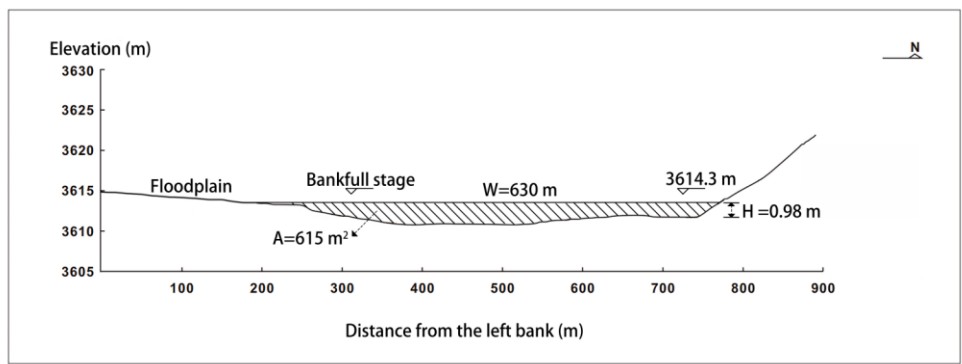

**Figure 7.** Calculated bankfull channel dimensions at N4 section.

Riverbed samples were collected mainly by vertical pipe borehole sampling. Two parallel boreholes were obtained at each location, with a distance of ≤2 m (by default, the sedimentary material and burial depth of these two boreholes are identical). Through the main borehole, stratigraphic information was obtained, the sediment sequence was delineated, and the layer to be dated was then selected according to the sediment lithology. Then, the sediments were obtained at the same burial depths in the secondary borehole for luminescence dating. After sediment acquisition, each end of the iron tubes was filled with cotton and wrapped with cling film, tin foil, and opaque tape to ensure that the samples did not undergo any further light exposure or moisture loss during transport.

2.4.2. Equivalent Dose ($D_e$) Determination

The luminescence dating instrument was a Risø TL/OSL DA-20 (Roskilde, Denmark, Risø centre). The blue light laser for stimulating quartz was $470 \pm 30$ mm, and the infrared laser (830 nm) was used for feldspar. Two 7.5 mm HoyaU-340 filters were placed in front of the photomultiplier, and the artificial β-radiation source was $^{90}Sr/^{90}Y$. All experimental steps were performed in a dark room under red light (central wavelength, ~$655 \pm 30$ nm). The procedures by Lai and Ou [29] were followed to extract quartz: 10% hydrochloric acid and 30% hydrogen peroxide were added to the samples to remove carbonates and organic matter, respectively. The coarse-grained fraction (90–125 μm) was extracted from all samples, except from N4-1-3, where the fraction of 38–63 μm was extracted based on the

grain size availability. To obtain pure quartz minerals, the 38–63 μm grains were soaked in 35% fluorosilicic acid for 2 weeks. The 90–125 μm grains were soaked in 40% hydrofluoric acid for 40 min, washed subsequently with 10% dilute hydrochloric acid and water, before being dried. Prior to the on-board testing, IR excitation experiments were performed on each sample to ensure that there was no feldspar contamination [30] and to avoid age underestimation [31]. If a significant signal was obtained, the samples were re-treated with fluorosilicic/hydrofluoric acid.

For equivalent dose ($D_e$) determination, the combination of the single aliquot regeneration (SAR) method [32] and the standard growth curve (SGC) method [33], and the SAR-SGC method were applied to determine the equivalent sample dose. Previous studies have shown that the combined SAR-SGC method can greatly reduce machine time while ensuring the accuracy of the dating results [22,34–36]. The experimental SAR parameters were set according to Lai [34]. For each sample, the $D_e$s and quartz growth curves of six aliquots were first measured using the SAR method, and an SGC was constructed using these six curves. Then, 10–12 aliquots were used to measure the natural signal (Lx) and the test dose (Tx) signal; their $D_e$s were obtained by matching the Lx/Tx values into the SGC. The final $D_e$ of the samples was taken as the arithmetic mean of all $D_e$s (including both the SAR and SGC methods).

### 2.4.3. Environmental Dose Rate Determination

To determine the environmental dose rate for the OSL samples, the content of radioactive elements U and Th were measured using inductively coupled plasma mass spectrometry (ICP-MS), and the content of K was obtained using inductively coupled plasma emission spectrometry (ICP-OES). As all samples were derived from fluvial sediments, the estimated water content of $20 \pm 10\%$ was employed for all samples. For quartz grains of 38–63 μm, the $\alpha$-efficiency was taken as $0.035 \pm 0.005$ [32]. Subsequently, the contribution of cosmic rays to the dose rate was calculated based on the altitude, geographical location, and sampling depth of each sample [37]. The dose rate for each sample was calculated according to the formula and parameters provided by Aitken [38].

### 2.5. Calculation of Bankfull Area

Bankfull channel geometry in an alluvial river is described by the channel width, cross-sectional area, and the corresponding mean depth at the bankfull level, and these bankfull dimensions are important parameters in river training works, flood control engineering and research on river channel evolution [39,40]. Bankfull geometry is closely associated with the concept of hydraulic geometry developed by Leopold and Maddock [41], which is usually represented in the form of at-a-station or downstream hydraulic geometry. Many hydraulic geometry relations have been proposed to describe the bankfull dimensions in stable or quasi-stable rivers, using empirically fitted power functions of a characteristic discharge or a controlled drainage area [42–44].

The cross-section of a typical alluvial river can usually be divided into two parts: the main trough and the beach. When the river runoff is small, the water level in the river is lower than the bankfull stage, and the water flow is all concentrated in the main tank for transportation. With the increase in runoff, the water level continues to rise. When the water level is equal to the floodplain surface of the river, the water level at this time is called the bankfull stage, and the corresponding discharge is called bankfull discharge. The cross-sectional area of the main trough below the bankfull stage is called the bankfull area.

The bankfull area can be obtained by using the bankfull stage line and the channel section lines to form a closed polyline, and using AutoCAD to automatically query the polyline's area (Figure 7). To obtain a bankfull area, the unit width of the river could be multiplied by the water depth of the corresponding location, and continuously accumulate the area.

# 3. Results and Discussions

## 3.1. Characterization of Sediments Layers in Boreholes in Each Sampling Section

The macroscopic sample characteristics were described in situ according to the soil, stratigraphic, and sedimentological methods. The N4 section results are shown in Figure 8 and Table 2.

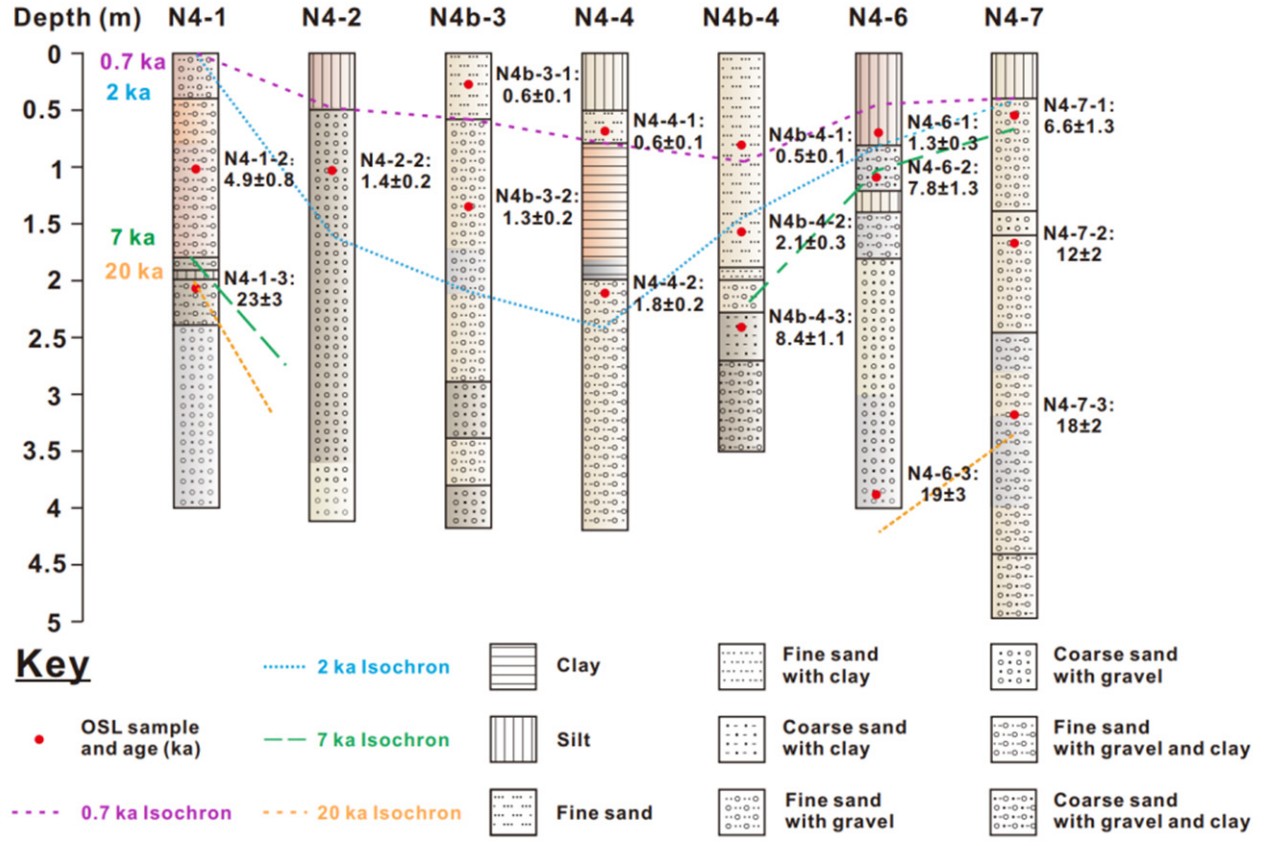

**Figure 8.** Drilled cores and dating results.

**Table 2.** Characterization of riverbed sediments at N4 profile of the Nangqian section along the Zhaqu River.

| Profiles | Location | Depth (cm) | Sedimentological Characteristics |
|---|---|---|---|
| N4-1 | Floodplain N: 32°8′29.8714″ S: 96°32′50.64817″ | 0–40 | Maroon fine sand with gravel content of 40% |
| | | 40–80 | Brownish red clay containing fine sand with gravel |
| | | 80–180 | Maroon clay containing fine sand with gravel |
| | | 180–240 | Gray-brown clay containing fine sand with pebble, gray-brown silty sand interlayer at 190–200 cm |
| | | 240–400 | Medium gray coarse sand with gravel |
| N4-2 | Mid-channel bar N: 32°8′34.13169″ S: 96°32′48.82664″ | 0–55 | Maroon silty sand to top soil |
| | | 55–200 | Gray-brown coarse sand with gravel |
| | | 200–360 | Gray-brown coarse sand with gravel |
| | | 360–410 | Brown-gray coarse sand with gravel |
| N4-3 | Mid-channel bar N: 32°8′36.25482″ S: 96°32′48.0900″ | 0–80 | Brown-yellow clay containing coarse sand with gravel |
| | | 80–200 | Gray sand and pebble layer |
| | | 200–240 | Yellow-brown coarse sand with gravel |
| | | 240–280 | Brown-yellow sand and pebble layer |

**Table 2.** *Cont.*

| Profiles | Location | Depth (cm) | Sedimentological Characteristics |
|---|---|---|---|
| N4-4 | Mid-channel bar N: 32°8′45.32052″ S: 96°32′51.76142″ | 0–55 55–80 80–185 185–200 200–420 | Yellow-brown silty sand with plant roots Yellow-brown fine sand Red-brown clay, hard plastic Black clay Yellow-brown clay containing fine sand |
| N4-5 | Mid-channel bar N: 32°8′45.32052″ S: 96°32′51.76142″ | 0–85 85–125 125–185 185–200 200–385 | Brown-yellow silty sand Medium gray medium sandy clay layer, gray-black clay formed in lacustrine environment, plastic, Brown-yellow medium sand Medium gray medium sandy clay layer, plastic Medium gray sand and gravel layer |
| N4-6 | Floodplain N: 32°8′49.05118″ S: 96°32′53.21237″ | 0–80 80–120 120–140 140–180 180–300 300–400 | Maroon silt Gray coarse sandy clay with gravel layer Brown-yellow silty sand with black and foul odor Dark gray silty sand with gravel Brown-gray coarse sand and gravel layer Dark gray coarse sand and gravel layer |
| N4-7 | Floodplain N: 32°8′50.37248″ S: 96°32′54.21672″ | 0–40 40–140 140–160 160–250 250–280 280–320 320–440 440–500 | Tawny silty sand Tawny fine sand with gravel Brown-yellow coarse sand with gravel Brown-yellow fine sand with gravel Gray clay containing fine sand with gravel and rock types $\Phi$ > 10 cm Brown-yellow clay containing fine sand with gravel Gray clay containing fine sand with gravel and rock types $\Phi$ > 10 cm Yellow-brown clay containing coarse sand with gravel |
| N4b-1 | Mid-channel bar N: 32° 8′30.64″ S: 96°32′58.46″ | 0–28 | Yellow-brown silty sand |
| N4b-2 | Floodplain N: 32°8′34.13169″ S: 96°32′48.82664″ | 0–80 80–210 210–280 280–390 | Yellow-brown sand layer Yellow-brown clay containing sand with gravel Brown-yellow silty sand with gravel Brown-yellow fine sand with gravel |
| N4b-3 | Floodplain N: 32°8′36.04″ S: 96°32′55.31038″ | 0–60 60–170 170–210 210–290 290–340 340–380 380–420 | Brown-yellow fine sand with residual plant roots Brown-yellow fine sand with gravel Gray fine sand with gravel Yellow-brown fine sand with gravel Brown-gray coarse sand and gravel layer Yellow-brown fine sand with gravel Brown-gray coarse sand and gravel layer |
| N4b-4 | Floodplain N: 32°8′48.12072″ S: 96°32′54.7075″ | 0–190 190–200 200–230 230–270 270–320 | Brown-yellow fine sand Yellow-brown silt-clay layer Yellow-brown fine sand with 5% gravel content, well rounding Gray-brown clay containing coarse sand Gray-brown clay containing coarse sand with gravel |

*3.2. OSL Dating Results*

The equivalent dose divided by dose rate, provided the length of time the minerals were buried (i.e., sample age [38]). Dating details are listed in Table 3, and ages are also labeled in Figure 8.

The growth and signal decay curves of two representative samples, N4-7-2 and N4-7-3, can be seen in Figure 9, where the red growth curves (Figure 9a,c) were obtained from averaging the other six growth curves of each sample and served as the sample SGC. Notably, the similar shapes of the growth curves of six aliquots indicate that the SAR-SGC method was applicable.

**Table 3.** Sample information, annual dose rate, and OSL ages: *a* and *b* are the number of aliquots measured using SAR and SGC methods, respectively.

| Sample Number | N4-1-2 | N4-1-3 | N4-2-2 | N4-4-1 | N4-4-2 | N4-6-1 | N4-6-2 | N4-6-4 | N4-7-1 | N4-7-2 | N4-7-3 | N4 b-3-1 | N4 b-3-2 | N4 b-4-1 | N4 b-4-2 | N4 b-4-3 |
|---|---|---|---|---|---|---|---|---|---|---|---|---|---|---|---|---|
| Grain size (μm) | 90–125 | 38–63 | 90–125 | 90–125 | 90–125 | 90–125 | 90–125 | 90–125 | 90–125 | 90–125 | 90–125 | 90–125 | 90–125 | 90–125 | 90–125 | 90–125 |
| Depth (m) | 0.96 | 1.95 | 1.05 | 0.67 | 2.13 | 0.74 | 1.05 | 3.78 | 0.52 | 1.51 | 3.03 | 0.26 | 1.33 | 0.74 | 1.59 | 2.28 |
| Measurement (No.) | 6 [a] + 10 [b] | 6 [a] + 10 [b] | 4 [a] + 11 [b] | 4 [a] + 10 [b] | 5 [a] + 11 [b] | 6 [a] + 12 [b] | 6 [a] + 12 [b] | 6 [a] + 11 [b] | 6 [a] + 12 [b] | 6 [a] + 12 [b] | 4 [a] + 9 [b] | 6 [a] + 10 [b] | 5 [a] + 10 [b] | 6 [a] + 8 [b] | 3 [a] + 11 [b] | 6 [a] + 12 [b] |
| K content (%) | 1.89 ± 0.19 | 1.53 ± 0.15 | 0.91 ± 0.09 | 1.18 ± 0.12 | 1.14 ± 0.11 | 1.2 ± 0.12 | 0.71 ± 0.07 | 0.75 ± 0.07 | 1.12 ± 0.11 | 0.92 ± 0.09 | 2.11 ± 0.21 | 1.26 ± 0.13 | 0.83 ± 0.08 | 0.98 ± 0.1 | 0.76 ± 0.08 | 0.82 ± 0.08 |
| Th content (ppm) | 9.62 ± 0.96 | 7.22 ± 0.72 | 5.79 ± 0.58 | 7.88 ± 0.79 | 7.2 ± 0.72 | 7.68 ± 0.77 | 4.53 ± 0.45 | 4.48 ± 0.45 | 7.31 ± 0.73 | 6.16 ± 0.62 | 15.16 ± 1.52 | 8.2 ± 0.82 | 5.29 ± 0.53 | 7.88 ± 0.79 | 4.86 ± 0.49 | 4.81 ± 0.48 |
| U content (ppm) | 2.97 ± 0.15 | 2.72 ± 0.14 | 2.09 ± 0.10 | 2.33 ± 0.12 | 2.17 ± 0.11 | 2.31 ± 0.12 | 1.76 ± 0.09 | 1.73 ± 0.09 | 2.23 ± 0.11 | 2.16 ± 0.11 | 2.78 ± 0.14 | 2.16 ± 0.11 | 2.10 ± 0.10 | 2.33 ± 0.12 | 1.73 ± 0.09 | 1.98 ± 0.10 |
| Water content (%) | 20 ± 10 | 20 ± 10 | 20 ± 10 | 20 ± 10 | 20 ± 10 | 20 ± 10 | 20 ± 10 | 20 ± 10 | 20 ± 10 | 20 ± 10 | 20 ± 10 | 20 ± 10 | 20 ± 10 | 20 ± 10 | 20 ± 10 | 20 ± 10 |
| $D_e$ (Gy) | 13.82 ± 1.1 | 58.6 ± 3.38 | 2.47 ± 0.23 | 1.25 ± 0.15 | 3.41 ± 0.18 | 2.74 ± 0.34 | 11.15 ± 0.85 | 25.52 ± 2.88 | 13.15 ± 0.42 | 21.21 ± 0.93 | 58.7 ± 1.8 | 1.28 ± 0.12 | 2.1 ± 0.21 | 0.92 ± 0.13 | 3.07 ± 0.3 | 12.6 ± 0.88 |
| Annual dose rate (Gy·ka$^{-1}$) | 2.84 ± 0.38 | 2.53 ± 0.3 | 1.71 ± 0.24 | 2.1 ± 0.34 | 1.93 ± 0.23 | 2.09 ± 0.32 | 1.43 ± 0.21 | 1.34 ± 0.15 | 2.01 ± 0.39 | 1.73 ± 0.21 | 3.18 ± 0.4 | 2.15 ± 0.44 | 1.61 ± 0.21 | 1.83 ± 0.29 | 1.48 ± 0.18 | 1.51 ± 0.17 |
| OSL age (ka) | 4.9 ± 0.8 | 23 ± 3 | 1.4 ± 0.2 | 0.6 ± 0.1 | 1.8 ± 0.2 | 1.3 ± 0.3 | 7.8 ± 1.3 | 19 ± 3 | 6.6 ± 1.3 | 12 ± 2 | 18 ± 2 | 0.6 ± 0.1 | 1.3 ± 0.2 | 0.5 ± 0.1 | 2.1 ± 0.3 | 8.4 ± 1.1 |

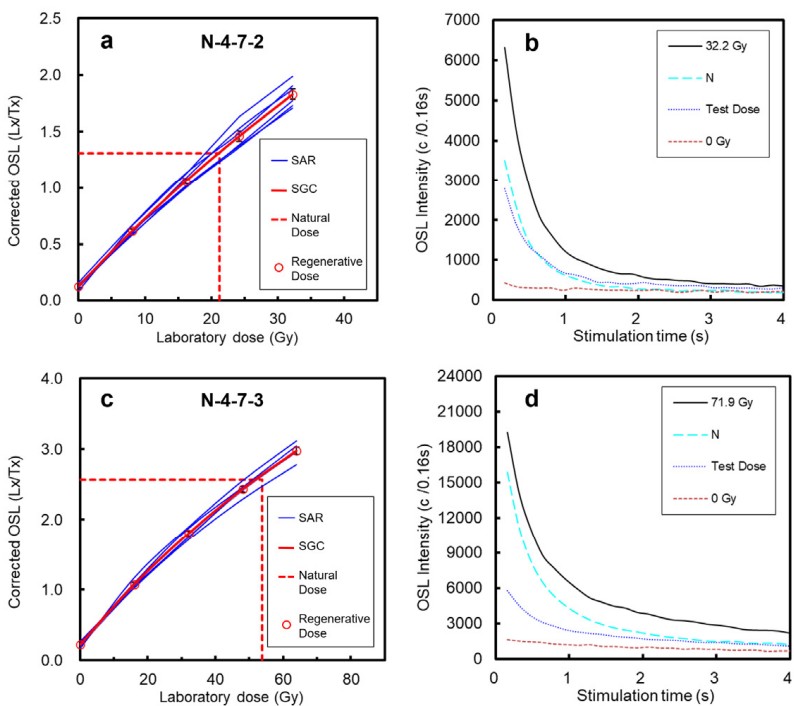

**Figure 9.** Growth curves (**a**,**c**) and decay curves (**b**,**d**) of two representative samples N4-7-2 and N4-7-3.

### 3.3. Historical Morphological Recovery for the Typical Cross-Sections and Their Response to Climate Change

In the source of the Lancang River, the two sets of samples, N4-4 and N4-4b were dated, labeled according to their depths, and the same chronological link was made on the topographic section map to obtain the historical section morphology (Figure 10).

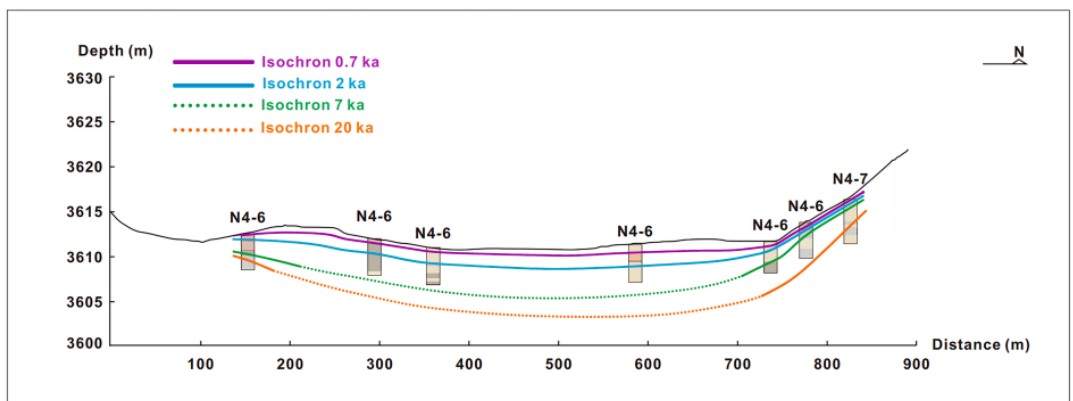

**Figure 10.** Historical morphological recovery for the N4 cross-section.

It was revealed that the river channel is relatively young, as the riverbed sediments were mainly ≤2 ka, thereby indicating that the river channel is significantly congested by the downstream Qinghai-Tibetan junction canyon—as are its sediments transport in the recent historical period. The recovered 0.7 ka and 2 ka sections were compared with the contemporary sections, and their general morphology was relatively consistent. Only the mainstream position differed (that from 0.7 ka is to the right), and the position of the deep flood (2 ka) section was consistent with the current one, which is roughly centered. When examining the elevation of the same age in the left bank stage to approximate the bankfull elevation during the historical period, it was revealed that the area of the 2 ka section was

notably large, although the difference between the 0.7 ka and contemporary sections was not significant.

Climate change research in China presented clear multi-timescale characteristics. According to the statistical analysis of China's climate over the past 5000 years, Zhu [45] believed that there were four warm and four cold periods, alternately.

From around 5 ka to 3 ka, it was a warm period. During most of this period, the annual average temperature was about 2 °C higher than the present, and the temperature of the coldest month was about 3 to 5 °C higher.

From 3 ka to 2.85 ka, there was a short, cold period.

From 2.8 ka to 2 ka, there was a second warm period.

From 2 ka~1.4 ka, there was a cold period.

From 1.4 ka to 1 ka, this was the third warm period.

From 1 ka to 0.8 ka, this was the third cold period, and the temperature is about 1 °C lower than that of modern times.

From 0.8 ka to 0.7 ka, it was the fourth warm period, but it is not as warm as 1.4 ka–1 ka. It gradually moved from the Huaihe River Basin to the south of the Yangtze River Basin.

The paleoclimatic evolution of the southeastern Tibetan Plateau, where the Lancang River originates, is broadly similar to the curve by Zhu [45]. For example, Zhang [46] found that the paleoclimate of the region went through seven stages of change: 5110–4570 before present (BP), the climate was generally warm and wet; 4570–2710 BP was marked by a decrease in both temperature and humidity; 2710–2360 BP, where the climate was the warmest and humid; 2360–1690 BP, where the climate shifted towards being more arid and cold; 1690–730 BP, both the temperature and humidity increased; 730–80 BP, where the climate was generally dry and cold; and from 80 BP to present, the temperature and humidity began to increase again. These findings are notably similar to those of the statistical analyses by Zhu [45]. Tan et al. [47] used high-resolution stalagmite $\delta^{18}O$ records to reveal an overall decreasing trend of the monsoonal precipitation over the last 2.3 ka on the southeastern Tibetan Plateau. The two most pronounced wet periods were identified from 1890–1670 and 1580–1440 BP; whereas the most pronounced dry period in the southeastern Tibetan Plateau occurred in the last 200 years. In addition, several interdecadal-scale wet periods were observed from 2120–2060 BP, 1110–960 BP, 840–810 BP, and 690–660 BP; as well as the interdecadal-scale dry periods from 1960–1915 BP, 1210–1110 BP, 790–705 BP, and 400–360 BP.

The recovery of the historical morphology of the river channel in the Nangqian section of the Lancang River source revealed that the channel was dominated by the morphology at the end of these historical warm periods, with significantly fewer sediments retained during the colder (and less cold) periods. The discontinuous cross-sectional morphology is indicative of a larger volume of water and more incoming sand during the warm periods, with more riverbed deposits. It can also be seen from Figure 10 that at the end of these historical warm periods in the Nangqian River section (e.g., 0.7 and 2 ka), the area of the section's main channel was notably larger than the present, as the water and sediments transport were large throughout the entirety of the warm periods, thereby shaping the corresponding wide and deep overflow sections. Furthermore, the terrace and river floodplain have not changed much from the present, indicating that the bankfull area in the warmer historical periods was larger than at present, and illustrated that its water and sediments transport capacity was stronger.

It is critical to calculate the bankfull areas for different geological times for a comparison of the river transportation capacities and the response of river morphology to climate change. After using the AutoCAD automatic query function for Figure 10, the following results are obtained: The current river is 630 m wide, 0.98 m at the bankfull stage, 615 m$^2$ in the bankfull area; the 0.7 ka river was 623 m wide, 1.27 m at the bankfull stage, 790 m$^2$ in the bankfull area; the 2 ka river is 618 m wide, 1.94 m at the bankfull stage, 1200 m$^2$ in the bankfull area. Overall, the bankfull area at 0.7 ka was ~1.28 times that of the current section, whereas the bankfull area at 2 ka was ~1.9 times greater than at present. Notably,

river width has not changed much because of the geomorphological constraints on both banks. In terms of sedimentation rates, the OSL age of the sediments at a depth of 2 m in the center of the Lancang River channel for the N4 section was ~2 ka, scilicet, and the sedimentation rate was ~1 mm/a.

The study here suggests that rivers may have a significant response to climate change, corroborating the use of paleoclimate reconstruction as a basis for predicting the fate of large Asian rivers in the context of present global climate change. It also offers strong support for recreating the scale of influence for ancient floods and weirs and is an effective tool for understanding river evolution patterns on millennial time scales.

## 4. Conclusions

Seven boreholes have been obtained following the cross-river transaction, and seventeen samples were dated by OSL. In order to investigate the response of river morphology to climate change, the paleochannel morphology was reconstructed based on the OSL ages and lithology, and the bankfull areas were calculated. Therefore, it is concluded that:

(1) The majority of riverbed sediments of the Nangqian reach in the Lancang River source were primarily channel-transported sediments deposited by the downstream canyon congestion since 2 ka. The general morphological cross-sections at 0.7 ka, 2 ka, and contemporary are consistent. The thalweg location of the 2 ka section was similar to the present, while that of the 0.7 ka section was to the right bank. However, the area of the 2 ka section was significantly larger than at present, while no such significance was observed between the areas at 0.7 ka and the present.

(2) The alluvial riverbed section, notably dominated by the accumulative process, was selected for OSL dating on the stratified riverbed units, such as river terraces and floodplains. Dating results allowed for the possibility of recovering the historical, generalized cross-sections.

(3) The larger areas, recorded at 0.7 ka and 2 ka, recovered from the river morphology, are comparable to the present (1.28 and 1.9 times larger, respectively), reflecting the strong water and sediments transport capacity of this river section during the historical warm periods.

(4) This research will not only provide a basis for an in-depth understanding of the hydrological change process of the Lancang River source in the historical period, but it will also provide a solid basis for the discussion of paleohydrology and paleoenvironment.

**Author Contributions:** Y.Z. organized the project; Y.Z., Q.S. and Z.L. (Zhongping Lai) conceptualized the paper; Y.Z., Y.G., Q.S. and Z.L. (Zhongping Lai) wrote the paper; Y.Z., Q.S., Y.G., X.Y., X.L., S.Z., Y.L., Z.L. (Zhijing Li) and Z.L. (Zhongping Lai) joined the fieldwork and sample collections and participated in discussions; Q.S. performed OSL dating and analyzed the data. All authors have read and agreed to the published version of the manuscript.

**Funding:** This study was supported by the Young Top-notch Talent Cultivation Program of Hubei Province (2021-10), the Central Public-interest Scientific Institution Basal Research Fund (CKSF2021743 &2021485), and the National Natural Science Foundation of China (U2240226&91647117).

**Institutional Review Board Statement:** Not applicable.

**Informed Consent Statement:** Not applicable.

**Data Availability Statement:** Not applicable.

**Conflicts of Interest:** The authors declare no conflict of interest.

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
