# Peer review of "Response of Channel Morphology to Climate Change over the Past 2000 Years Using Vertical Boreholes Analysis in Lancang River Headwater in Tibetan Plateau"

_water, doi:10.3390/w14101593_

Round 1
Reviewer 1 Report
This manuscript presents an interesting study of river bathymetry. The field measurements and data analysis are significant. The structure of the manuscript is acceptable. Some suggestions for improving the manuscript are listed below:
1) Add some details about the methods of data analysis to the abstract.
2) Delete the gaps between adjacent paragraphs.
3) Add longitude/latitude as axis titles to Figure 1.
4) Add a length scale to Figures 3 and 6. Change the panel labels of Figure 6 to (a), (b), (c) and (d).
5) Increase the font size of Figure 4 and use black/white format, as opposed to colour.
6) Make Tables 1, 2, and 3 concise. Consolidate the superscripts a and b in Table 3.
7) Add panel labels to Figure 8.
Author Response
The authors are grateful for the constructive comments from editors and reviewers, which are of great value for revision. Please see the attachment.
- Add some details about the methods of data analysis to the abstract.
Reply: Thanks. We have added details about bankfull cross-section area that can reflect climate change in the Abstract. The details of bankfull cross-section can be seen in the method part (Section 2.5).
- Delete the gaps between adjacent paragraphs.
Reply: Agree and done.
- Add longitude/latitude as axis titles to Figure 1.
Reply: Thanks and added.
- Add a length scale to Figures 3 and 6. Change the panel labels of Figure 6 to (a), (b), (c) and (d).
- Reply: We have added a length scale to Figure 6 and changed it’s panel labels. Figure 3 is a schematic diagram.
- Increase the font size of Figure 4 and use black/white format, as opposed to colour.
Reply: Thanks. We have modified it.
- Make Tables 1, 2, and 3 concise. Consolidate the superscripts a and b in Table 3.
Reply: Thanks. We have simplified the tables.
- Add panel labels to Figure 8.
Reply: Thanks. We have added panel labels and modified (a) and (c).
Special thanks to you for your good comments.

Reviewer 2 Report
Topic should be shorten and changed as it may be misleading now. As a cross-section I (geomorphologist) understand a channel cross profile. Maybe adding word “geological” or use of “vertical boreholes” would clarify the paper topic.
Although the Lancang River is mentioned in the title in fact the study were conducted in the Nangqian reach. This should be clear for a reader from the very beginning.
Introduction gives no sufficient background of the topic. In present form first part of the introduction is just a brief review of methods and conclusions in publications related to subject area. Second part is an overview of dating methods. No information about riverbed responce to the climate changes in global and local scale.
There is lack of clearly defined aim of the study in relation to results achieved by other scientists.
Tables 1, 2 and 3 and Fig. 7 and 8 illustrate rather results than methods. Also in the text methods and results are mixed. Please shift results to chapter 3.
In chapter 2.3.1 samples “collected from two sections, N4 and N4b” are mentioned but in Fig. 7 one can find 5 sections N4 and 2 sections N4b. Please clarify.
In chapter 3 the results are mixed with information from other publication so I suggest to change the “Results” to “Results and discussion” or the entire part describing climate changes should be shifted to the Introduction. Please, take the cultural differences into account. For a reader from outside of China it will be completely incomprehensible to define timeline in terms of dynastic periods.
Last three paragraphs of the chapter 3 must originate from completely different text. My impression is that the authors have not read the text before submission.
The text of a scientific article should tell the reader the complete history of research beginning from the background, the idea, through methods to the results and finally the conclusions. The reviewed text is chaotic as if the authors obtained the data but were unable to convey the idea of ​​the research to the reader. The research background is not well defined. The authors do not specify the value their research brings to the science. The text of the article needs to be organised and the content should be divided thematically and placed in appropriate chapters.
The article is not suitable for publication in its current form.
Author Response
The authors are grateful for the constructive comments from editors and reviewers, which are of great value for revision. We considered carefully each point when revising. Please see the attachment.
This paper descripted riverbed drilling and OSL dating of Lancang River headwaters, and then the river morphology and the caculated bankfull area were successfully restored. In generally, the paper provides new river morphology for restoring bankfull area of Lancang River headwaters, which would provide new prospective for restoring paleohydrology and paleoenvironment. I think this paper deserves to be published by journal of Water.
- Topic should be shorten and changed as it may be misleading now. Maybe adding word “geological” or use of “vertical boreholes” would clarify the paper topic.
Reply: Agree and thanks. We have changed the title to “Response of Channel Morphology to Climate Change Over the Past 2,000 Years Using Vertical Boreholes Analysis in Lancang River Headwaters in Tibetan Plateau”.
- Although the Lancang River is mentioned in the title in fact the study were conducted in the Nangqian reach. This should be clear for a reader from the very beginning.
Reply: Thanks for the comments and suggestions. We have moved this part about Nangqian reach from the third paragraph of the introduction to the first paragraph.
- Introduction gives no sufficient background of the topic. No information about riverbed responce to the climate changes in global and local scale.
Reply: Thanks. We have added some descriptions on the response of riverbeds to climate change. (The 6th paragraph of the introduction, in line 105-116.)
- There is lack of clearly defined aim of the study in relation to results achieved by other scientists.
Reply: Thanks. We have added the aim of study at the last paragraph of the introduction as follow (The 7th paragraphs of the introduction. In line 117-126. ) :
In the current study, boreholes were obtained following a cross-channel transaction, and sediments were dated by OSL, in order to restore the channel morphology and establish the rlationship between riverbed evolution and climate change in the Nangqian basin of the Lancang River headwater. The size of the bankfull cross-section area was calculated based on the reconstructed cross-section channel morphology and runoff. Sediments transport capacity was also estimated.
- Tables 1, 2 and 3 and Fig. 7 and 8 illustrate rather results than methods. Also in the text methods and results are mixed. Please shift results to chapter 3.
Reply: Thanks. We have moved them to chapter 3.
- In chapter 2.3.1 samples “collected from two sections, N4 and N4b” are mentioned but in Fig. 7 one can find 5 sections N4 and 2 sections N4b. Please clarify.
Reply: Thanks. In Figure 7 (Now it is Figure 8.), the numbering has a format such as "N4-1". "N4" means it comes from the "N4" section. "-1" means it is the "first borehole" of the section.
- In chapter 3 the results are mixed with information from other publication so I suggest to change the “Results” to “Results and discussion” or the entire part describing climate changes should be shifted to the Introduction. Please, take the cultural differences into account. For a reader from outside of China it will be completely incomprehensible to define timeline in terms of dynastic periods.
Reply: Thanks. We have changed “Results” to “Results and discussions” and deleted the content about dynastic periods in line 326-337.
- Last three paragraphs of the chapter 3 must originate from completely different text. My impression is that the authors have not read the text before submission.
Reply: Thanks, and very sorry for the mistake. We have proofread the whole of the manasript.
- The text of a scientific article should tell the reader the complete history of research beginning from the background, the idea, through methods to the results and finally the conclusions. The reviewed text is chaotic as if the authors obtained the data but were unable to convey the idea of the research to the reader. The research background is not well defined. The authors do not specify the value their research brings to the science. The text of the article needs to be organised and the content should be divided thematically and placed in appropriate chapters.
Reply: Thanks. We have added the research background (in line 117-126) and research methods (in line 259-282), and modified the conclusions (in line 404-406).
Special thanks to you for your good comments.

Reviewer 3 Report
Please revise the manuscript according to the attached file.

Author Response
The authors are grateful for the constructive comments from editors and reviewers, which are of great value for revision. We considered carefully each point when revising. Please see the attachment.
This paper descripted riverbed drilling and OSL dating of Lancang River headwaters, and then the river morphology and the caculated bankfull area were successfully restored. In generally, the paper provides new river morphology for restoring bankfull area of Lancang River headwaters, which would provide new prospective for restoring paleohydrology and paleoenvironment. I think this paper deserves to be published by journal of Water.
- Abstract: Reaction of riverbed to climate change was investigated through different sources like field surveys, drilling, etc. Did you also use satellite images to enhance your results?
Reply: Thanks. We found this particular reach based on satellite images. This is an accumulation reach, which is helpful for our research. We have added relevant content in line 45-46.
- Abstract: It is pointed to "sand transport capacity of river during warm period". Is the selected river bed consisted of only sand? In general, sediments of gravel or even boulder sizes may transport during floods.
Reply: Thanks for the comment. Your suggestion is correct. The river bed isn’t consist of only sand. We have changed "sand transport capacity" to "sediments transport capacity".
- The "Introduction" includes a proper literature review; however, at the end of that section the main aim of the paper was not described enough. Moreover, the author should specify the novelty of their study with respect to the previous ones.
Reply: Thanks. We have added them at the whole last two paragraphs of the introduction (in line 117-132).
- Figure 2: The legend is ambiguous. Are the numbers the topographic elevations? What is the unit? meter? Please complete the legend.
Reply: Thanks. We have completed the legend. The unit of Figure 2 is meters.
- Does Figure 3 belong to specific cross-section of the river? This figure is correct only in a section where sediment deposition exceeds the erosion rate. Please clarify in the manuscript.
Reply: Agree and thanks. You can see this special condition in line 171-172 in the first paragraph of Section 2.2.
- The section "4. Conclusions" is similar to a technical note and does not sound scientific. Please lay emphasis on the novelty of your study as well as new ideas and suggestions to solve or at least reduce the selected river problem.
Reply: Thanks for the suggestion. We have added some content in line 404-406 as part four of the conclusion as follow.
This research will not only provide a basis for in-depth understanding of the hydrological change process of the Lancang River source in the historical period, but also provide a solid basis for the discussion of paleohydrology and paleoenvironment.
Special thanks to you for your good comments.

Round 2
Reviewer 2 Report
Dear Authors,
the paper you submitted is in my opinion still not perfectly designed scientific paper, but it is acceptable in current form. I guess You are young scientist and there is still a lot of time for you to gain more experience.
I wish You a lot of successes on Your scientific path!
Best regards,
Reviewer